# A substrateless, flexible, and water-resistant organic light-emitting diode

Changmin Keum [1], Caroline Murawski[1,3], Emily Archer [1], Seonil Kwon[1], Andreas Mischok [1] & Malte C. Gather [1,2 ✉]

Despite widespread interest, ultrathin and highly flexible light-emitting devices that can be seamlessly integrated and used for flexible displays, wearables, and as bioimplants remain elusive. Organic light-emitting diodes (OLEDs) with µm-scale thickness and exceptional flexibility have been demonstrated but show insufficient stability in air and moist environments due to a lack of suitable encapsulation barriers. Here, we demonstrate an efficient and stable OLED with a total thickness of ≈ 12 µm that can be fully immersed in water or cell nutrient media for weeks without suffering substantial degradation. The active layers of the device are embedded between conformal barriers formed by alternating layers of parylene-C and metal oxides that are deposited through a low temperature chemical vapour process. These barriers also confer stability of the OLED to repeated bending and to extensive postprocessing, e.g. via reactive gas plasmas, organic solvents, and photolithography. This unprecedented robustness opens up a wide range of novel possibilities for ultrathin OLEDs.

[1] Organic Semiconductor Centre, SUPA, School of Physics and Astronomy, University of St Andrews, St Andrews, UK. [2] Centre for Nanobiophotonics, Department of Chemistry, University of Cologne, Köln, Germany. [3] Present address: Kurt-Schwabe-Institut für Mess- und Sensortechnik Meinsberg e.V., Waldheim, Germany. ✉email: mcg6@st-andrews.ac.uk

The unique properties of organic semiconductors—in particular their amorphous and mechanically pliable nature—have enabled a wide range of flexible optoelectronic devices, including organic light-emitting diodes (OLEDs)[1–3], organic solar cells[4–6], organic sensors[7], electronic skin[8], and neural devices[9,10]. The development of mechanically flexible OLEDs has inspired smartphones and TVs with curved displays and first products with simple foldable displays are now entering the market. Beyond their use in displays, flexible OLEDs enable a multitude of promising new applications in which conformal integration or resilience against mechanical deformation are essential, e.g., for wearable and biomedical devices[11–13]. In all these cases, an ultrathin form factor is highly desirable to reduce weight and volume, enable ultimate mechanical flexibility and conformability, and importantly, minimize mechanical strain in the device upon bending and folding.

Ultrathin OLEDs with impressive flexibility have been reported[11,14–19], but for such ultrathin devices stable operation under ambient or even aqueous conditions has not been achieved, with lifetimes in air still only in the range of tens of hours typically. The main challenge in manufacturing reliable flexible OLEDs is the extreme sensitivity of organic semiconductors to moisture and oxygen. To prevent rapid device degradation, a thin-film encapsulation (TFE) is required that is flexible yet provides a robust hermetic seal. Microscopic pinholes or microcracks that may form during deposition of the TFE or when bending the device result in rapid device failure. The situation is particularly challenging for future biomedical uses of OLEDs as these applications frequently require bio-implantation and thus resistance to aqueous environments.

Early studies demonstrated TFEs based on inorganic thin films[20,21], often formed by atomic layer deposition (ALD), which allows the deposition of conformal, densely packed, and hence pin-hole free metal oxide films[22,23]. Nanolaminates formed by alternating ultrathin layers of two different inorganic materials were found to improve encapsulation performance further[24–26]. As an extension of this concept and to improve compatibility with flexible substrates, inorganic-polymer multilayer structures have been proposed as TFE barriers[27,28], and such structures were indeed found to show promising barrier properties[29–33]. However, the flexible OLEDs reported in the literature so far either show poor stability under ambient conditions due to weak or non-existent TFE, or used relatively thick plastic substrates with embedded barriers. For commercial flexible displays, in addition to the substrate and TFE barriers, other functional layers such as a cover window are bonded into stacks using adhesives, yielding overall thicknesses of hundreds of μm, which has limited mechanical flexibility, has increased the weight and form factor and has added complexity to device manufacturing.

In this work, we demonstrate ultrathin, flexible, and efficient OLEDs that are resistant to air, water, various solvents, and reactive gas plasmas. The active layers of our OLEDs are sandwiched between two identical hybrid TFE barriers consisting of inorganic nanolaminates and parylene-C. As there is no need for a substrate, this approach leads to a total device thickness of ≈12 μm—similar to the typical thickness of cling film for food packaging. The symmetric, sandwich-like structure ensures that the active layers of the device are located in the neutral plane of the device, where they can flex without exposure to tensile or compressive stresses[34]. Parylene-C is an FDA-approved material with excellent biocompatibility that is widely used to coat biomedical devices[8,9,11,35]. Recently, parylene-C has been tested as TFE for OLEDs[10,16,36], but its moisture permeability is far too high[37] to provide effective protection for OLEDs when used on its own. In contrast, OLEDs protected by our flexible hybrid TFE barrier show no degradation in performance after more than 70 days in ambient air, are stable in water and cell culture media for at least two weeks, and tolerate repeated folding (e.g., 5000 cycles to a bending radius of 1.5 mm). Optical modeling of the active layers and the TFE barrier is used to enhance the light extraction efficiency. Optimized red-emitting OLEDs reach over 17% external quantum efficiency and over 40 lm W$^{-1}$ luminous efficacy. Additionally, we demonstrate that the TFE barrier can be tuned to function as a light scattering structure to further improve the light extraction efficiency.

## Results

**Device structure and characterization of flexible barriers.** Our flexible OLEDs are composed of two ≈6 μm thick TFE barrier films sandwiching the actual device, which consists of a semi-transparent metal anode, the active organic layers, and a highly reflective metal cathode (Fig. 1a). During processing, a carrier substrate is used, but the final device has no further components and thus is thin, bendable, and light weight (Fig. 1b). Each TFE barrier film consists of two pairs of Al$_2$O$_3$/ZrO$_2$ nanolaminates (N, 50 nm thick, deposited by ALD) and a parylene-C layer (P, 3 μm thick, deposited by chemical vapor deposition, CVD). For the active organic layers, we adopted a red phosphorescent p-i-n architecture as our testbed, but the concept described here should be compatible with most other state-of-the-art OLED stacks and was also tested for blue fluorescent OLEDs (Supplementary Fig. 1 and Supplementary Movie 1). Although our device is nominally a bottom-emitting design, the vertical symmetry of the thin, free-standing OLEDs means it can equally be used in top-emission by simply flipping the device around.

To investigate the encapsulation performance and optical properties of the TFE barrier, we tested three different configurations of the lower barrier by either (i) removing the nanolaminate on the side facing the OLED (P/N/P), (ii) replacing the nanolaminate with 50 nm of Al$_2$O$_3$ (P/N/P/A), or (iii) keeping the 50 nm Al$_2$O$_3$/ZrO$_2$ nanolaminate as described above (P/N/P/N). (We strongly expect that the trends seen for the different lower barrier configurations will be similar if the upper barrier was changed but this was not tested.) Figure 1c shows atomic force microscopy (AFM) of the surface of each of the three tested lower barrier layers and of a bare display-grade glass substrate. Compared with display glass, the parylene-C surface of the P/N/P barrier was relatively rough, but deposition of the Al$_2$O$_3$ layer or the Al$_2$O$_3$/ZrO$_2$ nanolaminate onto the parylene-C smoothened the surface (see Fig. 1c for root-mean-square roughness, $Rq$, of each sample). While the bare display glass was nearly 10-fold flatter than the bare parylene-C, deposition of the silver bottom electrode led to a significant increase in roughness on glass, most likely due to an island-like film growth. By comparison, the increase in roughness upon silver deposition was much less for the other samples, possibly due to better adhesion and reduced diffusion of metal atoms on their surface. All three tested barriers were highly transparent; their mean transmittance across the 400–800 nm range was 94.4%, 95.0%, and 92.0% for the P/N/P, P/N/P/A, and P/N/P/N sample, respectively. Due to the low thickness of the barrier layer and the refractive index contrast between the organic and inorganic layers, the transmission spectra showed pronounced thin-film interference (Fig. 1d; for refractive indices of used materials, Supplementary Fig. 2).

**Device efficiency and spectral characterization.** Optical modeling of the entire device stack was used to optimize the thickness of the charge transport layers in the OLED for maximum light outcoupling efficiency (Supplementary Fig. 3). The model predicts a series of efficiency maxima for the different orders of the optical microcavity that is formed by the device electrodes.

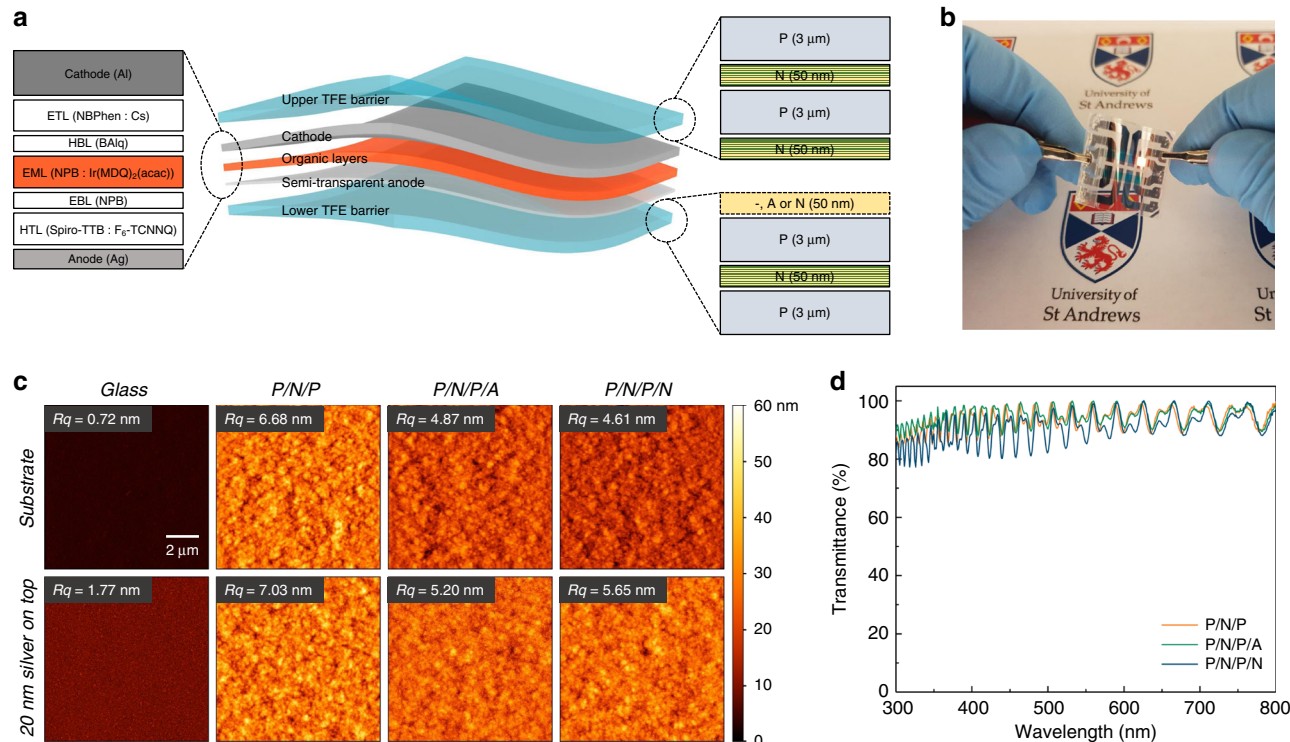

**Fig. 1 Flexible, substrateless OLEDs with water-resistant, hybrid TFE barriers. a** Schematic illustration of stack architecture with active layers in neutral plane (middle). Structure of the red-emitting p-i-n OLED stack used (left). Layer composition of TFE barriers (right). Barriers consist of parylene-C (P), $Al_2O_3$ (A), and a nanolaminate of alternating $Al_2O_3$ and $ZrO_2$ layers each with a nominal thickness of 3 nm (N). **b** Photograph of a flexible OLED with P/N/P/N lower TFE barrier, top-right pixel operated. **c** AFM topography images of the bare top surface of different lower TFE barrier layers and of display-grade glass (top row) and of the same samples after depositing a 20 nm-thick semi-transparent silver anode (bottom row). $Rq$, root-mean-square roughness across each image. **d** Transmittance spectra for each of the three lower TFE barriers.

Devices with first and second order cavities were fabricated and, in line with predictions of the model, first order devices reached the highest efficiency. However, statistical analysis showed that second order devices were significantly more stable electrically, with 10 to 100-fold lower leakage currents (defined as current at 1 V, Supplementary Fig. 4). Here, the thicker transport layers were presumably more effective in compensating the roughness of the underlying metal electrode and barrier layer. While roughness of the TFE barrier can likely be reduced by further optimization of the deposition process, for this study we focused on second order devices to minimize leakage current and obtain electrically stable devices.

Due to the microcavity effect discussed above and the additional thin-film interference from the TFE barrier, the electroluminescence (EL) spectra and angular emission characteristics of our devices were more complex than for conventional bottom-emitting OLEDs with glass substrates of millimeter-thickness. Compared with a reference OLED (same microcavity OLED stack deposited on a glass substrate with the top TFE barrier), the EL spectra of our flexible devices exhibited pronounced oscillations with wavelength (Fig. 2a and Supplementary Fig. 5), in line with the transmittance spectra of the TFE barrier layers discussed above. However, these spectral oscillations occurred across a narrow wavelength range and therefore did not impact color quality and visual perception. In addition, as expected for a microcavity device architecture, the peak emission wavelength blue-shifted with increasing viewing angle. Interestingly, the peak shift was least pronounced in the device with a P/N/P/N lower barrier, because the high-refractive index nanolaminate affects the phase change upon reflection at the silver bottom electrode. The microcavity architecture also led to a

substantial deviation of the angular distribution of emission intensity from the ideal Lambertian distribution (Supplementary Fig. 5). Again, the P/N/P/N device differed from the others; it emitted the highest radiant intensity in forward direction, whereas the maximum occurred at a viewing angle of 20–25° for the other OLEDs. Comparing all four devices, the transition from a super-Lambertian to a sub-Lambertian profile occurs at 32°, 47°, 50°, and 52° for P/N/P/N, P/N/P/A, P/N/P, and glass-based OLEDs, respectively.

Figure 2b shows the current density–voltage–luminance (*j–V–L*) characteristics of the P/N/P, P/N/P/A, and P/N/P/N devices and the glass reference OLED. In agreement with the differences in the angular emission distribution, the P/N/P/N OLED showed the highest luminance in forward direction, exceeding 5000 cd m$^{-2}$ at 4 V. Likewise, when computing the external quantum efficiency (EQE) from the *j–V–L* characteristics and assuming Lambertian distribution of emission intensity, the P/N/P/N structure reached substantially higher values than the other devices (Fig. 2c). However, when taking the angular emission characteristics of each device into account in the calculation, the EQE of all devices was relatively similar. The power efficacy of the devices was substantially larger when considering the actual angular emission characteristics than when assuming Lambertian distribution (Fig. 2d). We attribute this to the blueshift in EL with increasing viewing angle, which led to a substantial increase in luminous intensity at higher angles due to increased overlap with the photopic response of the eye (luminous intensity distribution, Supplementary Fig. 5).

Overall, we found that OLEDs deposited on and encapsulated by the hybrid TFE barriers developed here showed *j–V–L* characteristics and efficiency that were comparable to reference

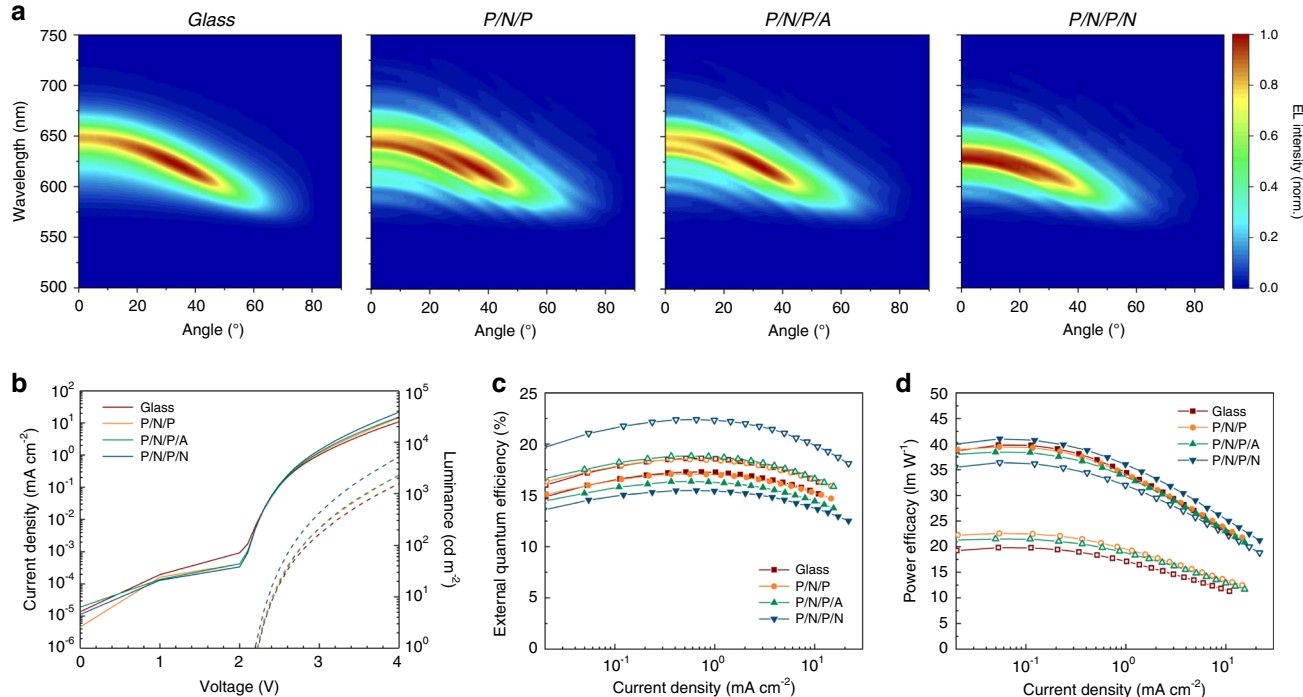

**Fig. 2 Angle-resolved EL spectra and efficiency characterization. a** 2D map of normalized, angle-resolved EL spectra of substrateless OLEDs with different lower TFE barriers and for a rigid reference device on display-grade glass. Recorded at constant current density of 7.5, 7.0, 6.2, and 3.1 mA cm$^{-2}$ for the glass, P/N/P, P/N/P/A, and P/N/P/N devices, respectively. **b** Current density (solid lines) and luminance (dashed lines) versus voltage for same devices. **c** External quantum efficiency and **d** power efficacy of OLEDs, calculated assuming a Lambertian emission profile (open symbols) and corrected for the actual angular emission profile (closed symbols).

devices produced on display grade glass. Furthermore, the absolute values of EQE and luminous efficacy were in line with earlier reports for similar phosphorescent p-i-n OLED stacks[38,39].

**Long-term stability**. Despite its hydrophobic nature, parylene-C alone is not able to prevent penetration of oxygen and moisture. OLEDs fabricated on lower barrier layers formed solely by parylene-C showed good performance while still on the glass carrier substrate, but they stopped functioning almost immediately after peeling them from the carrier (Supplementary Fig. 6). On the other hand, Al$_2$O$_3$/ZrO$_2$ nanolaminates provided some barrier function in air, even when used on their own[26]. However, when immersed into a liquid, e.g., cell culture medium, the bare Al$_2$O$_3$/ZrO$_2$ nanolaminate failed rapidly, which resulted in the detachment of the entire OLED from the substrate (Supplementary Fig. 7), thus illustrating the requirement for a hybrid approach.

Figure 3a shows the luminance–voltage characteristics of OLEDs with the three different lower TFE barriers after storage in air for different lengths of time. After 70 days in air, the luminance reached at 4 V reduced by 21.5% and 20.6% for the devices with P/N/P and P/N/P/A barrier, respectively. By contrast, the P/N/P/N device exhibited almost identical performance after 40 days and only showed a very moderate decrease in luminance (<10%, at 4 V) after 70 days of storage in air. This difference is further illustrated by the appearance of dark spots in the active pixel area of the P/N/P and P/N/P/A devices after extended storage in air while the P/N/P/N devices showed homogenous EL across the entire area even after 70 days (Fig. 3b). The operational lifetime of the flexible OLEDs under constant current driving (Supplementary Fig. 8) was consistent with the stability reported in the literature for the same emitter material and a similar p-i-n OLED stack[40].

We found that the flexible hybrid TFE barrier not only allowed extended storage of our devices in air, but also provided sufficient protection to store and operate the devices in deionized water, cell culture media and mild organic solvents like acetone and to expose them to conventional photoresists (Fig. 3c and Supplementary Fig. 9, Supplementary Movies 2 and 3). Here, the difference between the three different barrier structures was even stronger than in air (Fig. 3d). The luminance of the P/N/P device decreased to about 50% of its initial value after 150 h in deionized water. The P/N/P/A device remained relatively stable over the first 150 h but began to decay rapidly from this time onwards. By contrast, OLEDs with P/N/P/N barrier showed no significant loss in luminance when immersed into water for more than two weeks (350 h). We also tested the stability of OLEDs with the P/N/P/N barrier under 1× phosphate-buffered saline (PBS) solution, which is a frequently used buffer in cell and tissue culture applications, and under a cell culture medium (Fig. 3e and Supplementary Fig. 10). The PBS sample was kept at room temperature while the latter sample was additionally transferred into a cell culture incubator held at 37 °C. After 2 weeks, no degradation was observed for the device in contact with PBS and the current density decreased by only 10% for the sample under culture medium. These results illustrate that the flexible OLEDs developed here can be useful as bio-implantable devices.

**Bending stability**. To test the mechanical stability of our devices, we carried out a series of bending tests. First, the devices were bent with different radii of curvature, down to $r_b = 1$ mm, in both outward and inward direction (defined relative to the direction of light-emission as sketched in Fig. 4a). The current passing through the devices at a fixed voltage after 1 min of bending did not differ from its original value, regardless of the composition of the bottom TFE barrier (Fig. 4a). Figure 4b shows a photograph

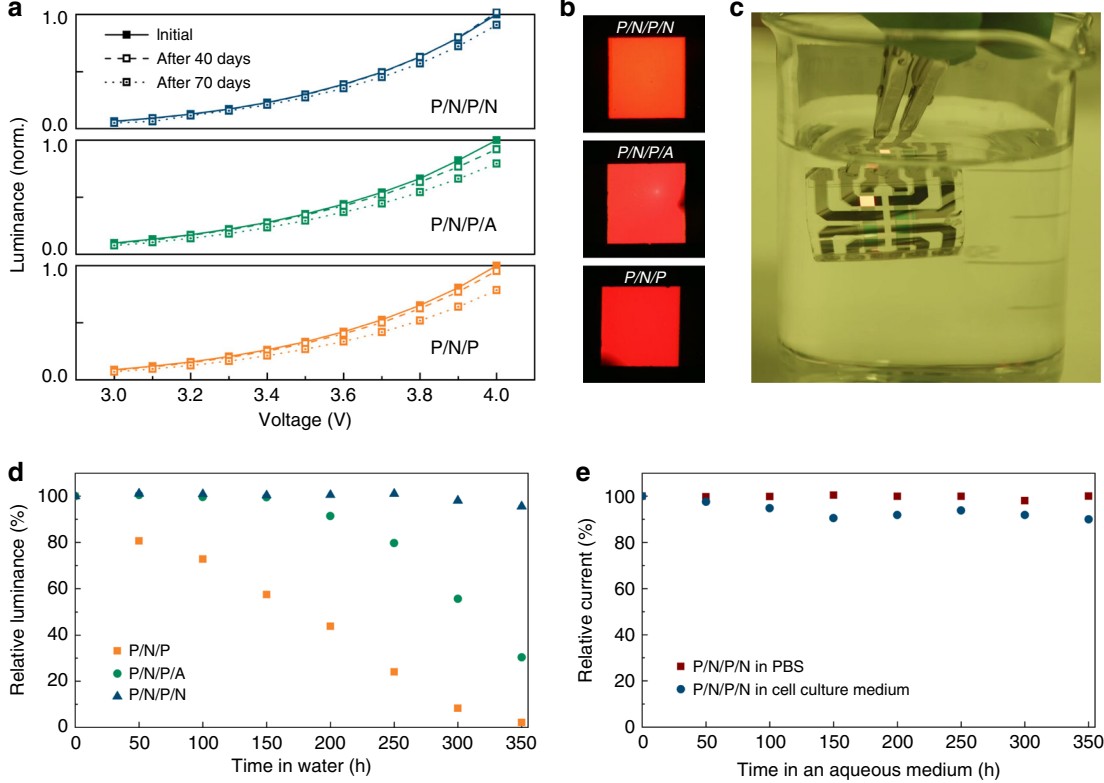

**Fig. 3 Stability of flexible OLEDs in air and under aqueous environments. a** Luminance–voltage characteristics of flexible OLEDs with different TFE barriers after storage in air for different lengths of time. Data normalized to initial luminance at 4 V for each device. **b** Photographs of operated pixels on flexible OLEDs after 70 days in air (pixel size, 2 mm × 2 mm; voltage, 2.5 V). **c** Photograph of P/N/P/N device operated in deionized water. **d** Change in luminance over time at fixed voltage (3.5 V) for flexible OLEDs with different TFE barriers stored and operated in deionized water. **e** Change in current density at fixed voltage (3.5 V) for P/N/P/N devices in 1× PBS solution (at room temperature) and in cell culture medium (at 37 °C and under an atmosphere of 5% $CO_2$ and 99% relative humidity).

of a more extreme case, in which an OLED is operated while being folded around the back of a razor blade ($r_b \approx 0.2$ mm). Supplementary Movies 4 and 5 show further demonstrations of device bending and twisting. The resilience of our devices to these extreme bending events is largely the result of their thin form factor; even for $r_b = 0.2$ mm, the strain at the surface of the TFE barrier is only $\varepsilon = T/2r_b = 3\%$ (where $T$ is the total device thickness). The symmetric design of our devices places the active layers and the adjacent nanolaminates in the neutral plane of the structure where bending causes neither compressive nor tensile stress[34]. By using parylene-C on the outside of the TFE barriers, the outermost metal oxide nanolaminate is only around 3 μm away from the neutral plane and thus experiences less than 1% strain for $r_b = 0.2$ mm.

Next, long-term mechanical stability was tested by 5000 repeated bending cycles at $r_b = 1.5$ mm using a device with a P/N/P/N barrier. The $j$–$V$–$L$ characteristics of the device remained unchanged during the entire test; in particular, there was no increase in leakage current (Fig. 4c). If present, such an increase would indicate the formation of microdefects. At a fixed operating voltage of 3.3 V (corresponding to a luminance >700 cd m$^{-2}$, higher than the typical requirement for commercial displays), the variation in current density and luminance over the course of the bending cycle test was less than 6% (Fig. 4d). These results confirm that OLEDs with the hybrid TFE barrier proposed here can provide stable and bright emission under extensive and frequent bending.

**In situ plasma etching of light outcoupling structures**. Optical simulations show that the relatively high refractive index of parylene-C ($n = 1.64$) and the nanolaminate ($n = 1.87$) has a two-

fold consequence for light outcoupling in our flexible OLEDs (Supplementary Fig. 11). First, due to increased Fresnel reflection at the air-device interface, outcoupling efficiency is reduced by about 5% relative to a reference device with a glass substrate. However, due to the low contrast in refractive index between the active layers of the OLED and the TFE barrier, waveguided modes that are mostly confined to the active layers in OLEDs with a glass substrate extend significantly into the TFE barrier. This leads to an approximately three-fold higher optical power in the TFE barrier of the flexible OLEDs than in the glass substrate of a reference device.

A wide range of light outcoupling structures have been described in the literature[41,42], including multiple light scattering schemes. For these structures to extract the light confined in waveguided modes, which in typical second-order OLEDs amounts to 30–50% of the total optical power generated[43], they usually have to be integrated close to the active layers, which can have a negative effect on electrical stability. However, in our case, because the waveguided modes extend into the TFE barrier, light outcoupling structures located at the device-air interface—and thus away from the active layers—could be used to access the waveguided modes. Our model predicts that if all light were extracted from the TFE barrier, the outcoupling efficiency would increase by around 60%.

The high stability and strong protection of our TFE barrier allowed us to trial a unique in situ method to create a light outcoupling structure directly within the outermost parylene-C layer of our flexible OLEDs. This was tested using a top-emitting OLED architecture (Fig. 5a). After depositing the upper barrier on the OLED stack, we exposed the top parylene-C layer to a

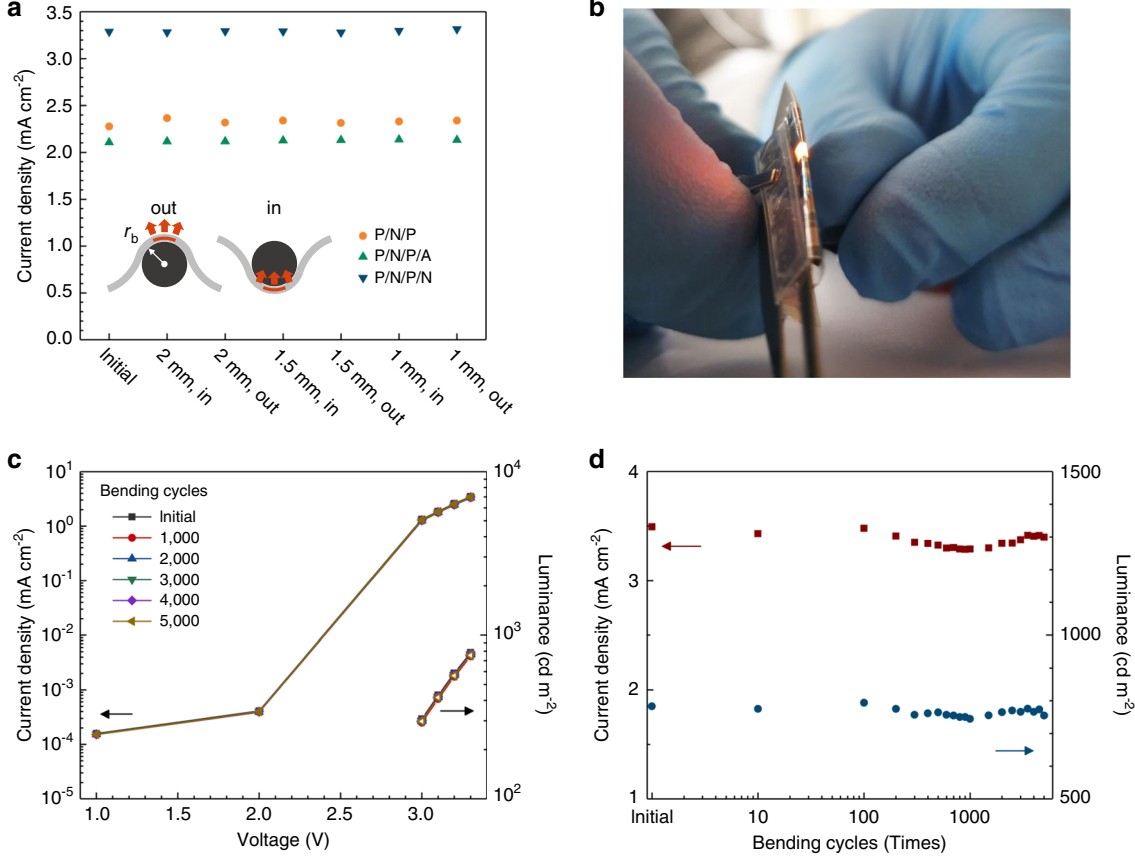

**Fig. 4 Stability of OLED performance under bending. a** Current density at 3.3 V after bending to increasingly smaller bending radii $r_b$. Inset illustrates definition of outward and inward bending with respect to the direction of light emission. **b** Photograph of flexible OLED folded around a razor blade. **c** Current density and luminance versus voltage for a P/N/P/N device, as fabricated and after thousands of repeated bending cycles. **d** Current density and luminance at 3.3 V versus number of applied bending cycles (bending radius of 1.5 mm).

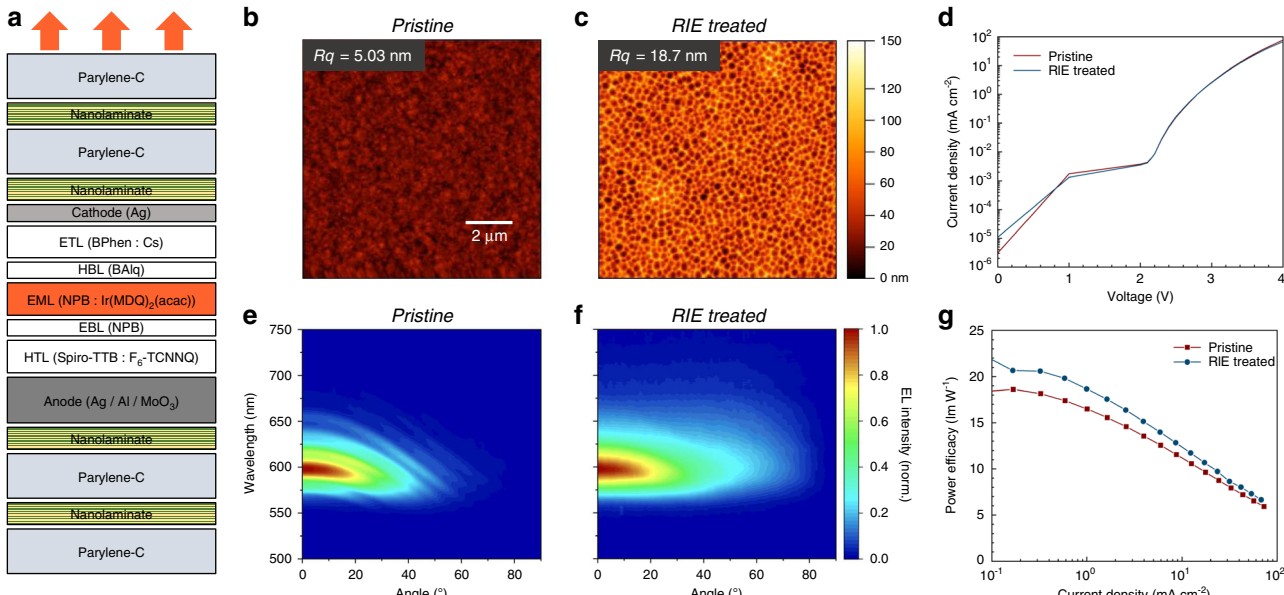

**Fig. 5 Roughening of the TFE barrier by plasma etching and effect on light outcoupling. a** Stack architecture of flexible top-emitting OLED with TFE barriers. **b** AFM topography image of a pristine parylene-C surface. **c** Topography of same surface after roughening by RIE on the same scale as in **b**. **d** Current density vs. voltage characteristics of pristine OLED and after RIE treatment. **e** 2D map of normalized, angle-resolved EL spectra for pristine OLED. **f** Same for RIE treated OLED. **g** Comparison of power efficacy between OLEDs with pristine and RIE treated TFE layers.

series of $O_2/SF_6$ gas plasma treatments using a reactive ion etching (RIE) system. This led to substantial roughening of the previously smooth surface, with the formation of deep pores with sub-μm diameters (Fig. 5b, c) but which importantly did not affect the electrical characteristics of the OLED (Fig. 5d). In addition, RIE treated OLEDs operated without issue for at least 900 h, exhibiting a slow and smooth decay in luminance over time, which indicates the absence of extrinsic sources of degradation (Supplementary Fig. 12).

As with the bottom-emitting OLEDs discussed above (Fig. 2a), the EL spectrum of a top-emitting OLED with pristine barrier showed strong oscillations and shifted to the blue with increasing viewing angle (Fig. 5e). By contrast, RIE treatment of the barrier removed the spectral oscillations and led to an angle independent EL spectrum, which we attribute to the presence of substantial surface scattering (Fig. 5f and Supplementary Fig. 12). In addition, the angular distribution of emission intensity was broadened compared to the pristine device and more closely resembled a Lambertian pattern (Supplementary Fig. 12). The power efficacy of the OLED with an RIE roughened outer surface was 13% higher than for the pristine device (at $1 \, \text{mA} \, \text{cm}^{-2}$, Fig. 5g). As the current–voltage characteristics were not affected by the RIE treatment, we conclude that this increase is due to improved light outcoupling. While the increase is smaller than the maximum possible value estimated by optical modeling, we expect that further optimization of the scattering strength will allow for a greater increase in light outcoupling efficiency.

## Discussion

Organic semiconductors degrade in the presence of hard radiation, high energy particles and high process temperatures. This makes the use of common high-throughput physical vapor and plasma-assisted deposition processes on top of OLED stacks problematic and thus renders them less preferable for the manufacture of TFE barriers on OLEDs. In addition, for a TFE barrier to be efficient, it has to be free of pinholes, and thus it should be tolerant to inhomogeneities in the topography of the underlying layer, e.g., from particle contamination. Through the use of metal oxide ALD and CVD of parylene-C, we combined two low-temperature, chemically driven deposition modalities that both have highly conformal coating characteristics. The majority of the barrier consists of parylene-C, which is widely used in the electronics industry and can be deposited at low cost and with high throughput. The in situ polymerization of the para-xylylene derivative that yields the parylene-C polymer film is thermally initiated, does not require a solvent or crosslinker, and generally forms no by-products, thus rendering it ideally suited for the deposition on highly sensitive organic electronics. Likewise, the reaction of metal oxide ALD precursors proceeds in a quantitative and highly controlled manner and forms very dense films. In the past, ALD has been associated with limited throughput because each deposition cycle only yields a single atomic layer and takes of order 10 s to complete in a lab-scale reactor. However, through reactor optimization, cycle times have been reduced to the sub-second scale, and emerging modalities such as spatial ALD have demonstrated even 100-fold improvements in speed over conventional ALD, thus largely invalidating these concerns[44]. Besides the individual advantages of the ALD nanolaminate and the CVD parylene-C, the combination of both appears to be particularly attractive for a TFE barrier that is to be used in moist environments. We attribute this in part to the hydrophobic nature of parylene-C, consistent with earlier reports showing that hydrophobic $SiO_2$-polymer composites perform better than polyvinyl alcohol[17]. In addition, the non-contact, gas-phase deposition of parylene-C minimizes the mechanical stress applied to the thin

underlying nanolaminate, which should help to conserve its integrity. Parylene-C has been reported to withstand temperatures of 80 °C in air for around 100,000 h; other variants like parylene-HT tolerate temperatures up to 350 °C[37]. The metal oxide nanolaminate is expected to show even higher temperature stability. Future testing should explore the thermal stability of the hybrid TFE system in detail.

Emerging commercial flexible displays use macroscopic packaging and have thicknesses of hundreds of micrometers. They are generally based on a pre-produced flexible substrate, such as a poly-imide film or lately an ultrathin glass, have a pre-produced cover window and often employ additional barrier films. All of these are typically laminated together using adhesives. While this approach allows for good device stability, it adds weight and reduces mechanical flexibility. In addition, the transition from the gas/vacuum phase deposition of OLED and TFE to the solution phase deposition of adhesives adds complexity and requires careful control of particle contamination on the OLED facing side of the outer layers. The in situ fabrication of TFE barrier and OLED in a substrateless fashion reported here avoids many of these challenges.

The robustness, extreme form factor and mechanical flexibility of our substrateless OLEDs opens up possibilities for a number of future uses. For instance, they might be laminated or affixed onto or into work surfaces, packaging and clothing, where they could be used as self-emissive indicators and labels that do not add significant weight and volume to the product. The stability of our devices under high humidity and in water makes them suited for wearable applications requiring skin-contact and for use as implants in biomedical research; for the latter we have a particular interest in using them for targeted photostimulation and optical recording of neuronal activity via optogenetics[45,46]. Finally, the robustness of the TFE barrier to harsh process conditions, including reactive gas plasmas, photoresists, and organic solvents, is unprecedented for ultrathin OLEDs. It provides attractive opportunities for post-processing, e.g., defining microstructures on OLEDs via dry-etching into their surface or performing lithographic patterning of additional layers deposited onto the device.

## Methods

**Fabrication and characterization of flexible TFE barriers.** Display grade glass substrates (Eagle XG, Corning) were thoroughly cleaned in acetone, isopropanol, and oxygen plasma. Parylene-C (diX C, KISCO) and thin-film oxide layers were deposited using a parylene coater (Labcoater 2, SCS) and an ALD reactor (Savannah S200, Ultratech), respectively, with both coaters connected to a common nitrogen filled glovebox. The parylene-C powder was vaporized at 130–140 °C and the gaseous dimer was pyrolyzed into a monomer at 690 °C. The polymeric films of parylene-C were then formed on the devices in the main vacuum chamber of the parylene coating system which was kept at room temperature and at a base pressure of <25 mTorr. Nanolaminate layers of alternating $Al_2O_3$ and $ZrO_2$ sublayers were deposited following the recipes described in our previous report[26]. In brief, 20 cycles of a 15 ms Trimethylaluminum (TMA) pulse/10 s $N_2$ purge/15 ms $H_2O$ pulse/10 s $N_2$ purge and 28 cycles of a 300 ms tetrakis(dimethylamino)zirconium (TDMAZr) pulse/7 s $N_2$ purge/30 ms $H_2O$ pulse/7 s $N_2$ purge were performed to produce 3 nm thick $Al_2O_3$ and $ZrO_2$ sublayers, respectively. The TDMAZr precursor was heated to 75 °C; the TMA and $H_2O$ cylinders were maintained at room temperature. The process temperature and base pressure of the ALD reactor were 80 °C and 0.1 Torr, respectively.

AFM topography maps were acquired under ambient conditions using contact mode AFM (FlexAFM system, Nanosurf) using tipped cantilevers (ContAl-G) and a force set point of 10 nN. Transmittance spectra were recorded for samples prepared on quartz disks using a UV–Vis spectrophotometer (Cary 300, Varian).

**Device fabrication.** Red p-i-n OLEDs were fabricated via thermal evaporation at a base pressure lower than $3 \times 10^{-7}$ mbar (EvoVac, Angstrom Engineering). The layer sequence from bottom to top was Ag (20 nm) as anode/Spiro-TTB doped with $F_6$-TCNNQ at 4 wt% (50 nm for first order cavity, 220 nm for second order) as hole transport layer (HTL)/NPB (10 nm) as electron blocking layer (EBL)/NPB doped with $Ir(MDQ)_2(acac)$ at 10 wt% (20 nm) as emissive layer (EML)/BAlq (10 nm) as hole blocking layer (HBL)/NBPhen doped with 3 wt% of Cs (60 nm for

the first order, 80 nm for second order) as electron transport layer (ETL)/Al (100 nm) as cathode. For top-emitting OLEDs, a multi-layered stack of Al (40 nm), Ag (60 nm) and $MoO_3$ (1 nm) was used as anode[47], Spiro-TTB doped with 4 wt% $F_6$-TCNNQ (190 nm) as HTL, BPhen doped with 3 wt% Cs (70 nm) as ETL, and Ag (20 nm) as cathode. The other layers were unchanged from the bottom-emitting configuration. All organic materials were purchased from Lumtec. An active pixel area of 2 mm × 2 mm was defined by shadow masking of the anode and cathode contact, with in situ mask exchange under high vacuum. After deposition of the OLED stack, devices were immediately protected by the upper TFE barrier. Devices were transferred to the ALD reactor through a nitrogen filled glovebox.

The acronyms of the materials used in the OLED stacks are as follows. Spiro-TTB: 2,2',7,7'-tetrakis-(N,N-di-methylphenylamino)-9,9'-spiro-bifluorene, $F_6$-TCNNQ: 2,2'-(perfluoronaphthalene-2,6-diylidene)dimalononitrile, NPB: N,N'-bis (naphthalen-1-yl)-N,N'-bis(phenyl)-benzidine, Ir(MDQ)$_2$(acac): (2-methyldibenzo [f,h]quinoxaline)(acetylacetonate) iridium(III), BAlq: bis(2-methyl-8-quinolinolate)-4-(phenylphenolato) aluminum, NBPhen: 2,9-dinaphthalen-2-yl-4,7-diphenyl-1,10-phenanthroline, BPhen: 4,7-diphenyl-1,10-phenanthroline.

**OLED characterization.** j–V–L characteristics were recorded using a source-measurement unit (2400 SourceMeter, Keithley Instruments) and a calibrated silicon photodiode. Angle-resolved EL spectra were acquired in steps of 1° using a custom-built automated goniometer setup equipped with a fiber-coupled spectrometer (Maya LSL, OceanOptics) operating the OLEDs in constant current mode. Efficiencies were calculated by both assuming a Lambertian emission profile and by taking the measured angular emission profiles into account. Variations in device performance in air or aqueous environments were measured by monitoring the relative luminance and current over time at a specific voltage using a source-measurement unit (2450 SourceMeter, Keithley Instruments) and a silicon photodetector (PDA100A2, ThorLabs). Operational lifetime was measured under constant current operation (M6000, McScience). For bending tests, devices were repeatedly flexed around fixed circular metal rods of different radii. A removable custom-built supporting frame was used for some of the electrical characterization to avoid damage to contact pads (Supplementary Fig. 13).

**Plasma etching of light outcoupling structures.** The outmost parylene-C layer of complete top-emitting OLEDs was roughened by exposure to an $O_2$ and $SF_6$ gas plasma in a custom-built RIE system. Under the process conditions used (power 150 W, $O_2$ gas flow rate 50 sccm, $SF_6$ gas flow rate 5 sccm) the etch rate was 350 nm min$^{-1}$. After etching away 2.5 µm of parylene-C, a further nanolaminate layer (30 nm) and another parylene-C layer (0.5 µm) were deposited, and a further 0.35 µm of parylene-C were etched. This protocol was found to maximize the light scattering effect, but further optimizations can likely be made.

**Optical simulations.** Optical simulations were based on the transfer matrix formalism described in Ref. [48]. Optical constants used in the simulation were measured by variable angle spectroscopic ellipsometry (M-2000DI, J.A. Woollam Co.) with each thin film layer deposited on silicon substrates. The multilayered oxide nanolaminate was approximated as a monolithic layer and its effective refractive index was measured to be an intermediate between $Al_2O_3$ and $ZrO_2$. This approximation was in good agreement with calculations for the actual multilayer nanolaminate structure (Supplementary Fig. 14).

## Data availability
Research data supporting this publication is available at https://doi.org/10.17630/ada14451-b64c-4aaf-8ed1-915116e8ec5f.

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

## Acknowledgements

This research was financially supported from the Leverhulme Trust (RPG-2017-231), the EPSRC NSF-CBET lead agency agreement (EP/R010595/1, 1706207), the DARPA NESD programme (N66001-17-C-4012) and the RS Macdonald Charitable Trust. C.K. acknowledges support from the Basic Science Research Program through the National Research Foundation of Korea (NRF) funded by the Ministry of Education (2017R1A6A3A03012331). C.M. acknowledges funding from the European Commission through a Marie Skłodowska Curie individual fellowship (703387). A.M. acknowledges funding through an individual fellowship of the Deutsche Forschungsgemeinschaft (404587082). M.C.G. acknowledges funding from the Alexander von Humboldt Stiftung (Humboldt-Professorship).

## Author contributions

C.K. and M.C.G. initiated the project, designed the experiments, and prepared the manuscript. C.K. fabricated and characterized the thin-films and OLED devices, performed AFM measurements, and analysed all data. C.M. and C.K. performed stability tests. E.A. and C.K. designed and built the goniometer measurement setup and measured angle-resolved spectra. S.K. and C.K. performed dry-etching and the related measurements. A.M. and C.K. performed optical simulations. All authors discussed the data and contributed to the manuscript.

## Competing interests

The authors declare no competing interests.
