## [Peer Review File · Nature Communications]

Reviewers' Comments:

Reviewer #1:

Remarks to the Author:

The manuscript addresses a critical area, which is the practical encapsulation of flexible OLEDs under immersion in water and saline media. I am commenting on the revision of the paper specifically.

While there has been significant past work in the use of laminates to provide a flexible, transparent sealing layer, the authors offer a unique fabrication process which improves the inter-layer adhesion and reduces the risk of delamination upon immersion in the liquids, which has plagued air-stable encapsulation schemes upon water exposure.

In addition to a thorough study of the performance, there are two major contributions of the work. A major contribution of the manuscript continues to be the excellent analysis of the extraction provided by the surface roughness on the external surface. This is well demonstrated and analyzed in the manuscript. The modeling of the optical extraction path is well presented and thoroughly analyzed. The second contribution is the use of a unique process for deposition of the inorganic layer which improves the interlayer adhesion with parylene (which is extremely permeable and inert) while retaining excellent performance for moisture permeation resistance under water (which is perhaps 100x more challenging than exposure to water vapor in air).

No further revision of the text is needed - the prior art is well referenced and structured, and the text is well presented and edited.

The photographs and graphics in the supplementary information are absolutely beautiful and remain a strength of the paper. These graphics present several attractive possibilities for cover art.

Reviewer #2:

Remarks to the Author:

The manuscript, entitled "A substrateless, flexible, and water-resistant organic light-emitting diode" by Gather group, reported highly durable, efficient, and foldable OLEDs by employing nanolaminates/parylene-C multibarrier encapsulation systems without typically thick substrates. Despite tremendous progress in the OLED technology, there is still a room to advance the OLED performance further particularly in terms of reliability in the harsh environments and foldability. The authors team demonstrated exceptional ultrathin OLEDs in all aspects of EL performance, mechanical properties, and durability. The systematic optical design/analyses and outcoupling enhancement strategy for the high-index TFE devices were appropriately implemented, and a complete experimental set for characterizations of TFE-introduced OLED devices was highly impressive. Therefore, this reviewer would like to strongly recommend this work to be published in this high-profile journal, Nature Communications, but after clarifying the following minor issues.

1. The authors demonstrated the high barrier property at the Al₂O₃-ZrO₂ nanolaminates/parylene-C multilayer TFE system. According to the ref. #17 (Energy Environ. Sci. 12, 1878-1889 (2019)), the Al₂O₃-ZnO nanolaminates/SiO₂-polymer composite TFE could also exhibit the remarkable barrier property that enabled OLED or OPV devices to work stably even in the detergent-added water over a couple of weeks. It would be highly convincing the readers of the present concept if the authors could provide insight into the high-performance TFE systems.
2. Please explain why Al₂O₃-ZrO₂ nanolaminates-only TFE could not support the high barrier property in the liquid? Were they dissolved?
3. For the upper TFE, is there any reason why the laminates were preferentially deposited before parylene-C? Please provide the brief description for parylene-C first cases (e.g. OLED/P/N/P and OLED/P/N/P/N/P).
4. The details of deposition process for the parylene-C layer needs to be clarified in the Methods

section (e.g. initiation temperature, post-annealing process, etc.) to fully resolve the concerns of the post-process environments as described in the Discussion section.

Reviewer #3:

Remarks to the Author:

The authors presented some interesting work with some eye-catching results and demos in laymen term, such as 12um thick OLED foil, substrate less, light emission emerged in water and 70 days without degradation. However, the work does not represent an exciting breakthrough by expert standard in the field. In term of thinness, flexible displays of 10um thickness, which had not only the OLED light emission but also driving thin-film transistors, was demonstrated over a decade ago. The substrate-less claim is also less of a surprise as all the flexible display panels such as being employed in foldable phones are made on a carrier substrate and subsequently peeled off, just as the authors did for their barrier film which in this case serves as the substrate. The parylene-C and ALD metal oxides hybrid barrier layer is hardly an innovation, as alternating organic/inorganic encapsulation is the industrial standard technology and mostly the depositions are low temperature processes. The most unprofessional presentation in this paper is the barrier property of their encapsulation film. The key property of WVTR (water vapor transmission rate) for the barrier film was not given, instead the authors used descriptions of emerging in water and left in air for 70 days. For a commercial OLED display panel, the standard requirement for encapsulation is to have the WVTR below 10^{-6} g/m²/day, which is equivalent to a spoon drop of water on an area of football field over a month period. With such high barrier property, leaving OLED in water or leaving in air (unknow humidity) for 70 days is just a piece of cake. The really rigorous environmental stability qualification for display panels is to leave the display device in 85% humidity at 85C temperature for over 2000 hours, the so-called double 85 test. As for the flexibility, 5000 bending cycles is a good number, but considering that the flexible display in a foldable phone has to pass 200k bending tests, 5000 bending is not qualified as brilliant. I am not saying that the paper is not publishable. It is just not particularly innovative to be published in the Nature Communications.

Point-by-point response to the reviewer comments

Reviewer #1:

The manuscript addresses a critical area, which is the practical encapsulation of flexible OLEDs under immersion in water and saline media. I am commenting on the revision of the paper specifically. While there has been significant past work in the use of laminates to provide a flexible, transparent sealing layer, the authors offer a unique fabrication process which improves the inter-layer adhesion and reduces the risk of delamination upon immersion in the liquids, which has plagued air-stable encapsulation schemes upon water exposure. In addition to a thorough study of the performance, there are two major contributions of the work. A major contribution of the manuscript continues to be the excellent analysis of the extraction provided by the surface roughness on the external surface. This is well demonstrated and analyzed in the manuscript. The modeling of the optical extraction path is well presented and thoroughly analyzed. The second contribution is the use of a unique process for deposition of the inorganic layer which improves the interlayer adhesion with parylene (which is extremely permeable and inert) while retaining excellent performance for moisture permeation resistance under water (which is perhaps 100x more challenging than exposure to water vapor in air). No further revision of the text is needed - the prior art is well referenced and structured, and the text is well presented and edited. The photographs and graphics in the supplementary information are absolutely beautiful and remain a strength of the paper. These graphics present several attractive possibilities for cover art.

>> We thank Reviewer #1 for the very positive review and were excited to read that he/she recommends publication of the manuscript in the current form.

Reviewer #2:

The manuscript, entitled "A substrateless, flexible, and water-resistant organic light-emitting diode" by Gather group, reported highly durable, efficient, and fordable OLEDs by employing nanolaminates/parylene-C multibarrier encapsulation systems without typically thick substrates. Despite tremendous progress in the OLED technology, there is still a room to advance the OLED performance further particularly in terms of reliability in the harsh

environments and foldability. The authors team demonstrated exceptional ultrathin OLEDs in all aspects of EL performance, mechanical properties, and durability. The systematic optical design/analyses and outcoupling enhancement strategy for the high-index TFE devices were appropriately implemented, and a complete experimental set for characterizations of TFE-introduced OLED devices was highly impressive. Therefore, this reviewer would like to strongly recommend this work to be published in this high-profile journal, Nature Communications, but after clarifying the following minor issues.

>> We thank Reviewer #2 for the positive overall review. Our point-by-point response to the minor concerns raised is provided below.

1. The authors demonstrated the high barrier property at the Al₂O₃-ZrO₂ nanolaminates/parylene-C multilayer TFE system. According to the ref. #17 (Energy Environ. Sci. 12, 1878-1889 (2019)), the Al₂O₃-ZnO nanolaminates/SiO₂-polymer composite TFE could also exhibit the remarkable barrier property that enabled OLED or OPV devices to work stably even in the detergent-added water over a couple of weeks. It would be highly convincing the readers of the present concept if the authors could provide insight into the high-performance TFE systems.

>> We agree that Ref. 17 also showed great barrier performance over a month although we understand that the effective immersion time was 10 min every 7 days. The authors of this study hypothesize that Al₂O₃ layers prepared using ALD can chemically react with the hydroxyl group of water. They then find that overcoating the Al₂O₃ with a hydrophobic SiO₂-polymer composite provides good barrier performance, possibly because of its hydrophobic nature. A similar effect may explain the high stability in our nanolaminate / parylene-C stack. Further research would be required to confirm this, but this is beyond the scope of our present study. However, in the Discussion section of our revised manuscript, we now mention this possibility in the context of Ref. 17 and trust that this addresses the referee's concern.

New text on p. 16

“Besides the individual advantages of the ALD nanolaminate and the CVD parylene-C, the combination of both appears to be particularly attractive for a TFE barrier that is to be used in moist environments. We attribute this in part to the hydrophobic nature of parylene-C, consistent with earlier reports showing that hydrophobic SiO₂-polymer composites perform

better than polyvinyl alcohol¹⁷. In addition, the non-contact, gas-phase deposition of parylene-C minimizes the mechanical stress applied to the thin underlying nanolaminate, which should help to conserve its integrity.”

2. Please explain why Al₂O₃-ZrO₂ nanolaminates-only TFE could not support the high barrier property in the liquid? Were they dissolved?

>> We think it is unlikely that the Al₂O₃-ZrO₂ nanolaminates dissolve in water or the other liquids we tested. Instead we believe that the device rapidly delaminates (as discussed in Supplementary Fig. 7). As mentioned above, according to Ref. 17, Al₂O₃ prepared using ALD may react with water which would lead to a change in volume relative to the pristine Al₂O₃ film. This volume change likely leads to local defects (cracks, pinholes), through which water can quickly penetrate given the low overall thickness of the nanolaminate. For OLEDs encapsulated only with a nanolaminate, this will lead to water reaching the organic layers, which then causes device failure. Even prior to chemical degradation of the organic layers, at least the TFE barrier and cathode appear to detach from the carrier substrate. As we also expect poor mechanical stability for a substrateless device without parylene-C (which would have a total thickness <1µm), we have not studied the stability of the nanolaminates-only TFE any further. To clarify this aspect about the nanolaminate-only TFE without distracting from the main topic of the hybrid TFE, we have added a further explanation to the caption of Supplementary Fig. 7.

Additional text for caption to Supplementary Figure 7

“While the reason for rapid failure of the nanolaminate-only TFE is not fully clear, it has been suggest that Al₂O₃ prepared using ALD reacts with water which in turn may lead to a change in volume relative to the pristine Al₂O₃ film. This volume change then would likely lead to local defects (cracks, pinholes), through which water can quickly penetrate given the low overall thickness of the nanolaminate. As we expect poor mechanical stability for a substrateless device without parylene-C (which would have a total thickness <1 µm), the stability of the nanolaminates-only TFE was not investigated further.”

3. For the upper TFE, is there any reason why the laminates were preferentially deposited before parylene-C? Please provide the brief description for parylene-C first cases (e.g. OLED/P/N/P and OLED/P/N/P/N/P).

>> We employed an upper TFE with N/P/N/P structure (with the P-layer facing outside) to make the device fully symmetric when paired with the P/N/P/N lower barrier (the primary structure used in our study) and thus ensure that the OLED layers are located in the neutral plane.

In terms of barrier performance, as shown in Figure 3 of the main text, the P/N/P lower barrier is not as efficient as P/N/P/N structure, and we very much expect the same result for the upper TFE. Given the logistical complexity of comparing different upper barrier layers, we feel that confirming this is beyond the scope of the present study.

Generally, in polymer-inorganic hybrid barrier systems, one of the main functions of the polymer layers is to decouple defects in adjacent inorganic barrier layers, which then leads to elongated permeation paths for moisture and oxygen. In this regard, having an additional parylene-C layer directly on top of the OLED stack would not improve the encapsulation performance substantially, given that the barrier property of parylene-C on its own is not particularly high (Supplementary Figure 6).

In addition, it is useful to place the nanolaminate layers as close as possible to the neutral plane of the device in order to minimize mechanical stress in these layers upon bending.

Finally, as explained in response to points 1 and 2 above, for use in moist or aqueous environments it may be beneficial to have the hydrophobic parylene-C as the outer layer to prevent a detrimental chemical reaction with the Al_2O_3 surface.

We have included most of the above points in our revised manuscript:

- Page 5: “(We strongly expect that the trends seen for the different lower barrier configurations will be similar if the upper barrier was changed but this was not tested.)”
- Page 12: “By using parylene-C on the outside of the TFE barriers, the outermost metal oxide nanolaminate is only around 3 μm away from the neutral plane and thus experiences less than 1% strain for $r_b = 0.2$ mm.”
- Page 16: “Besides the individual advantages of the ALD nanolaminate and the CVD parylene-C, the combination of both appears to be particularly attractive for a TFE barrier that is to be used in moist environments. We attribute this in part to the hydrophobic nature of parylene-C, consistent with earlier reports...”

4. *The details of deposition process for the parylene-C layer needs to be clarified in the Methods section (e.g. initiation temperature, post-annealing process, etc.) to fully resolve the concerns of the post-process environments as described in the Discussion section.*

>> We thank the reviewer for pointing us to this omission. We now added details on the parameters for the deposition of parylene-C to the Methods section. We stress that although the temperature of the pyrolysis chamber increases up to 690 °C, this does not affect the OLED devices as the deposition chamber containing the OLEDs is maintained at room temperature throughout the process.

New text on p. 18

“The parylene-C powder was vaporized at 130 ~ 140 °C and the gaseous dimer was pyrolyzed into a monomer at 690 °C. The polymeric films of parylene-C were then formed on the devices in the main vacuum chamber of the parylene coating system which was kept at room temperature and at a base pressure of < 25 mTorr.”

Reviewer #3

The authors presented some interesting work with some eye-catching results and demos in laymen term, such as 12um thick OLED foil, substrate less, light emission emerged in water and 70 days without degradation. However, the work does not represent an exciting breakthrough by expert standard in the field. In term of thinness, flexible displays of 10um thickness, which had not only the OLED light emission but also driving thin-film transistors, was demonstrated over a decade ago. The substrate-less claim is also less of a surprise as all the flexible display panels such as being employed in foldable phones are made on a carrier substrate and subsequently peeled off, just as the authors did for their barrier film which in this case serves as the substrate. The parylene-C and ALD metal oxides hybrid barrier layer is hardly an innovation, as alternating organic/inorganic encapsulation is the industrial standard technology and mostly the depositions are low temperature processes. The most unprofessional presentation in this paper is the barrier property of their encapsulation film. The key property of WVTR (water vapor transmission rate) for the barrier film was not given, instead the authors used descriptions of emerging in water and left in air for 70 days. For a commercial OLED display panel, the standard requirement for encapsulation is to have the WVTR below 10⁻⁶

g/m²/day, which is equivalent to a spoon drop of water on an area of football field over a month period. With such high barrier property, leaving OLED in water or leaving in air (unknown humidity) for 70 days is just a piece of cake. The really rigorous environmental stability qualification for display panels is to leave the display device in 85% humidity at 85C temperature for over 2000 hours, the so-called double 85 test. As for the flexibility, 5000 bending cycles is a good number, but considering that the flexible display in a foldable phone has to pass 200k bending tests, 5000 bending is not qualified as brilliant. I am not saying that the paper is not publishable. It is just not particularly innovative to be published in the Nature Communications.

>> Considering that the two other reviewers are very excited about the work, one even recommending publication without any changes, we find the harsh dismissal of our work by Reviewer #3 surprising. We understand that each reviewer may have different views, but we feel that the arguments raised by Reviewer #3 are mostly unjustified and in many instances not relevant to our study. Given Reviewer #3 makes a number of very general points, we first share some general observations before addressing each point individually.

The process reported here can be carried out in research labs, whereas the state-of-the-art claimed by the reviewer refers to industrial, large volume processes. Such industrial processes are (a) not described in the published literature and thus remain inaccessible to the community, and (b) cannot be performed at smaller scale in academic research labs, preventing their use and adaptation for novel, out-of-the-box applications of OLEDs.

In addition, although impressive encapsulation performance is achieved in commercial products today, these devices differ greatly from what is reported here, in both structure and function.

Many commercial OLED displays are indeed flexible on some level, either to conform to a curved cover glass (older Samsung Galaxy models; LG OLED TV) or recently in a few cases to be 'foldable' (Samsung Galaxy Fold). However, to our knowledge these displays use a pre-produced polyimide (PI) film or lately ultrathin glass as their substrate and in addition have a cover window material laminated onto the device with an adhesive. The stacks of flexible display packaging including a substrate, a cover window, adhesives, etc. are typically 100s of μm thick, which limits mechanical flexibility and adds weight but makes it much simpler to achieve good stability compared to the ultra-thin devices reported here.

By contrast, we show here how the entire structure can be made without using any pre-produced substrate or cover material. This approach makes it considerably easier to (a) obtain a perfectly symmetric structure that keeps the OLED in the neutral plane and (b) produce a very thin structure as no adhesive layers are required to bond the different parts of the device. We discussed this important difference between our devices and those produced by industry and reported in earlier literature in the introduction of our original manuscript and have now made the discussion more explicit.

Modified text on p. 3

“...the flexible OLEDs reported in the literature so far either show poor stability under ambient conditions due to weak or non-existent TFE, or used relatively thick plastic substrates with embedded barriers. For commercial flexible display, in addition to the substrate and TFE barriers, other functional layers such as a cover window are bonded into stacks using adhesives, yielding overall thicknesses of hundreds of μm , which has limited mechanical flexibility, has increased the weight and form factor and has added complexity to device manufacturing.”

Considering the above points, we feel that the terminology of a “substrateless OLED” is justified and accurate. Furthermore, we believe that demonstrating and quantifying good stability of the entire device in water, acetone, gas plasma and upon photoresist exposure is scientifically important to illustrate how flexible OLEDs can be used in areas that have not been considered so far.

Following this general statement, we would like to reply to the individual points raised by the referee as follows.

In term of thinness, flexible displays of 10 μm thickness, which had not only the OLED light emission but also driving thin-film transistors, was demonstrated over a decade ago.

>> While a number of very thin devices with different levels of integration have been reported over the years, earlier studies did not report levels of environmental stability comparable to the OLEDs reported in our study. In our view, integration with thin-film transistors, which typically have a thickness of only a few hundreds of nm, is not a big additional challenge in terms of developing ultrathin electronics with μm scale thickness. In fact, the majority of research papers in the field of flexible electronics focus on individual devices rather than on

integrating them with each other and leave the more highly integrated prototypes to industrial players. In the introduction to our manuscript, we clearly state that “Ultrathin OLEDs with impressive flexibility have been reported” but then point out that like the 10 µm thick flexible displays the referee mentions, such devices have lacked environmental stability, let alone being compatible with the harsh environments tested in our study.

The substrate-less claim is also less of a surprise as all the flexible display panels such as being employed in foldable phones are made on a carrier substrate and subsequently peeled off, just as the authors did for their barrier film which in this case serves as the substrate.

>> As mentioned above, commercial flexible display panels are generally based on a flexible substrate, such as a PI film or an ultrathin glass, and additionally contain further barrier films and adhesives. Our devices use no separate, pre-produced films and thus our approach avoids challenges like finding suitable adhesives and controlling particle contamination on the OLED facing side of the cover window. To clarify this point to readers who may have similar concerns we have added a new paragraph to the Discussion section of our manuscript.

New text on p. 16:

“Emerging commercial flexible displays use macroscopic packaging and have thicknesses of hundreds of micrometres. They are generally based on a pre-produced flexible substrate, such as a poly-imide film or lately an ultrathin glass, have a pre-produced cover window and often employ additional barrier films. All of these are typically laminated together using adhesives. While this approach allows for good device stability, it adds weight and reduces mechanical flexibility. In addition, the transition from the gas / vacuum phase deposition of OLED and TFE to the solution phase deposition of adhesives adds complexity and requires careful control of particle contamination on the OLED facing side of the outer layers. The in-situ fabrication of TFE barrier and OLED in a substrateless fashion reported here avoids many of these challenges.”

The parylene-C and ALD metal oxides hybrid barrier layer is hardly an innovation, as alternating organic/inorganic encapsulation is the industrial standard technology and mostly the depositions are low temperature processes.

>> We agree that metal oxides/organic hybrid barriers have been widely used and are regarded as one of the most promising TFE technologies for OLEDs. (We clearly acknowledge the success of organic/inorganic barriers in the introduction of our manuscript, “As an extension of this concept and to improve compatibility with flexible substrates, inorganic-polymer multilayer structures have been proposed as TFE barriers^{27,28} and such structures were indeed found to show promising barrier properties²⁹⁻³³.”)

However, our manuscript presents substantial advances in the composition, characterization, and application of alternating organic/inorganic encapsulation. The combination of parylene-C and the Al₂O₃/ZrO₂ nanolaminate and its stability under harsh environmental conditions have not been explored so far. In addition, to our knowledge there is no detailed analysis of the optical properties of alternating organic/inorganic TFE barrier, in particular when used in an ultrathin device. In this study, we report that the alternating and relatively high refractive index of such a TFE barrier has important implications for thin film interference and enables significant increases in light outcoupling compared to a glass substrate.

Little technical detail is available in the open literature on the material combinations used in industry, but we understand that e.g. the commercial Vitex/Barix® system uses reactively sputtered Al₂O₃ (high energy, potential sputter damage to organic layers, non-conformal coating unlike ALD) and a polyacrylate organic layer deposited by flash evaporation and subsequent UV curing (again high energy, potential UV damage and not intrinsically conformal). Other reports discuss solution-phase deposition of the organic component (e.g., Ref. 17 and Ref. 31 from our manuscript, or Sun, L. et al. *ACS Appl. Mater. Interfaces* **11**, 43425–43432 (2019), Duan, Y. et al. *Org. Electron.* **15**, 1936 (2014)) which can be low-cost and conformal but risks introducing particle defects and may lead to solvent induced degradation. Using parylene-C, with its adsorption-initiated polymerization at room temperature, combines a number of advantages as discussed in detail in our manuscript.

We believe that the specific advantages of the ALD nanolaminate / CVD parylene-C combination are discussed in sufficient detail in the manuscript and well summarized in the first paragraph of the Discussion section. We have therefore not made further changes in response to this point.

The most unprofessional presentation in this paper is the barrier property of their encapsulation film. The key property of WVTR (water vapor transmission rate) for the barrier film was not given, instead the authors used descriptions of emerging in water and left in air

for 70 days. For a commercial OLED display panel, the standard requirement for encapsulation is to have the WVTR below 10^{-6} g/m²/day, which is equivalent to a spoon drop of water on an area of football field over a month period. With such high barrier property, leaving OLED in water or leaving in air (unknown humidity) for 70 days is just a piece of cake.

>> We are of course aware that the WVTR is a widely used measure of barrier performance and that a WVTR below 10^{-6} g/m²/day is frequently quoted as a requirement for OLEDs. Unfortunately, the measurement sensitivity of standardized WVTR tests such as ASTM E96 and ASTM F1249 is limited to a range much higher than 10^{-6} g/m²/day. Instead, in the OLED community, WVTR is commonly measured by various electrical and optical calcium tests. However, there is still no standardized measurement protocol, and there is concern that Ca tests do not always provide a reliable WVTR value and that results depend strongly on the design of the test (see e.g., Klumbies, H. et al. *Rev. Sci. Instrum.* **85**, 2014–2017 (2014) and Nehm, F. et al. *Rev. Sci. Instrum.* **86**, 126110 (2015)). For instance, different groups perform Ca tests at different temperatures and humidity conditions and WVTR values are often measured for an arbitrary barrier thickness and then compared without normalizing for barrier thickness. In addition, the measurement sensitivity can vary by several orders of magnitude depending on environmental conditions (M. D. Kemp et al. *Rev. Sci. Instrum.* **84**, 025109 (2013)).

We are therefore concerned that WVTR values measured under non-standardized test conditions might mislead readers. In addition, developing a robust protocol to measure WVTR for the harsh environmental conditions studied in our work is beyond the scope of this paper. Instead, we believe it is more helpful to directly quantify the stability of state-of-the-art OLEDs in water and under the other relevant conditions.

We also like to point out that a WVTR value obtained under a specific condition has limited predictive value for the stability of actual devices under different conditions. For example, the WVTR of a 130 nm thick Al₂O₃/ZrO₂ nanolaminate was measured to be 4.7×10^{-5} g/m²/day at 70% RH and 70 °C by J. Meyer et al. (*Adv. Mater.* **21**, 1845-1849 (2009); Ref. 24; c.f. table below). The authors then estimate a room temperature WVTR for this same barrier of 5×10^{-7} g/m²/day by taking the activation energy of water vapor transmission through the barrier into account. The nanolaminates of our hybrid TFE barriers are deposited using a recipe that is based on Meyer et al. paper, and so we expect the WVTR of each of our Al₂O₃/ZrO₂ nanolaminates on their own to be in a similar range already. However, as shown in Supplementary Figure 7, Al₂O₃/ZrO₂ nanolaminates on their own fail within seconds when in contact with even a small droplet of water.

Table 1. Permeation rates for water and oxygen of Al₂O₃ layers and Al₂O₃/ZrO₂ nanolaminates. Environmental conditions: 70% RH and 70 °C.

Barrier layer (thickness [nm])		Permeation rate for water [g m ⁻² day ⁻¹]	Permeation rate for oxygen [cm ³ m ⁻² day ⁻¹]
Al ₂ O ₃	(100)	3.5×10^{-4}	1.2×10^{-1}
Al ₂ O ₃ + ZrO ₂	(100)	6.4×10^{-5}	2.1×10^{-2}
Al ₂ O ₃	(130)	9.9×10^{-5}	3.3×10^{-2}
Al ₂ O ₃ + ZrO ₂	(130)	4.7×10^{-5}	1.6×10^{-2}

Ref. 24, Adv. Mater. 21, 1845-1849 (2009)

Another example summarized in the table below shows that under certain conditions the WVTR of TFE barriers increases substantially after immersion in water (Ref. 17, Energy Environ. Sci. 12, 1878-1889 (2019)).

Table 4 The change in WVTR of 3 dyads encapsulation barriers according to various capping polymer layers and dipping storage conditions.

Polymer	Before dipping	After 1 day	After 3 days	After 7 days
PVA	$(6.65 \pm 0.89) \times 10^{-5}$ g/m ² /day	N/A	N/A	N/A
P _{low,SiO2}	$(2.04 \pm 0.47) \times 10^{-5}$ g/m ² /day	$(5.09 \pm 0.66) \times 10^{-4}$ g/m ² /day	$(1.09 \pm 0.32) \times 10^{-3}$ g/m ² /day	N/A
P _{high,SiO2}	$(1.18 \pm 0.44) \times 10^{-5}$ g/m ² /day	$(4.85 \pm 0.49) \times 10^{-5}$ g/m ² /day	$(1.10 \pm 0.41) \times 10^{-4}$ g/m ² /day	$(3.44 \pm 0.45) \times 10^{-4}$ g/m ² /day

^a These are statistical values of average and standard deviation obtained from 9 WVTR samples.

Ref. 17, Energy Environ. Sci. 12, 1878-1889 (2019)

In summary, even if a TFE stack shows excellent barrier performance with a WVTR of 10⁻⁶ g/m²/day or better, this will not guarantee that it can protect an operational OLED immersed in water. We expect that a vast majority of the OLED community will agree that achieving stability of highly efficient, state-of-the-art OLEDs under water with a 6 μm thin TFE represents a major breakthrough and is not “a piece of cake”.

The really rigorous environmental stability qualification for display panels is to leave the display device in 85% humidity at 85C temperature for over 2000 hours, the so-called double 85 test.

>> We agree that temperature and humidity testing is crucial in industry to evaluate the stability of display panels and many other electrical components and systems. When testing TFE barriers for OLED encapsulation at elevated temperatures, care must be taken to disentangle extrinsic degradation (i.e. failure of the barrier) and intrinsic degradation (e.g. due to limited thermal stability of the organic materials used in the OLED stack). The development of

temperature stable OLED materials is a research field of its own and many of the more stable materials are proprietary. A high temperature test is therefore beyond the scope of our study. To clarify that this is an aspect that will need to be studied further in the future, we have added a brief statement on temperature stability to the Discussion of our revised manuscript.

New text on p. 16:

“Parylene-C has been reported to withstand temperatures of 80 °C in air for around 100,000 hours; other variants like parylene-HT tolerate temperatures up to 350 °C³⁷. The metal oxide nanolaminate is expected to show even higher temperature stability. Future testing should explore the thermal stability of the hybrid TFE system in detail.”

Regarding humidity, we do not see how testing at 85% relative humidity (the protocols state under ‘non-condensing’ conditions) would be more rigorous than immersing the entire device in water.

Finally, we point out that the double 85 test suggested by the referee is not typically shown in the literature on OLED encapsulation. (In fact, as stated in the introduction of our manuscript, papers reporting “stable” flexible electronics, including several published in Nature Communications, achieve much lower stability than our devices, even under ambient conditions.)

As for the flexibility, 5000 bending cycles is a good number, but considering that the flexible display in a foldable phone has to pass 200k bending tests, 5000 bending is not qualified as brilliant.

>> Again, we feel it is important to compare like for like. We do not claim that at this stage our novel ultrathin OLEDs meet the industrial standard for displays with macroscopic thickness (100s of μm) that comprise multiple films which are held together by adhesives. Achieving 5000 bending cycles is in itself an impressive demonstration of mechanical flexibility and reliability. The fact that the performance of the tested OLEDs did not drop significantly over the 5000 bending cycles indicates that they might well survive a much larger number of cycles than we were able to test. We believe that we have objectively described the test we have performed and that the reader will not benefit from a more extended discussion of our results. Therefore, no revision was made in response to this comment.

Reviewers' Comments:

Reviewer #1:

None

Reviewer #2:

Remarks to the Author:

With the satisfactory responses arising from the authors, my previous concern is completely resolved. I could find the other reviewers' comments and related authors' responses, in which there was the considerable conflict in between. Although both are definitely reasonable, I believe the more persuasive arguments are toward the authors' side. Therefore, I would like to strongly recommend the revised manuscript for publication in Nature Communications without further revision.

Reviewer #3:

Remarks to the Author:

I recognize some of the authors' arguments. The industry do things differently from academics. They cannot compare like for like. The foldable phones have to pass much tougher tests in order not to upset the market and customers.

In term of innovations reported in this paper, putting an OLED film in water while it emitted light has been demonstrated by Holst Center years ago. The parylene/Al₂O₃ combination of multilayer barrier film was also reported before (Ref.36). The work which I think is innovative is to use the barrier film alone for OLED layers deposition. In this way, the OLED can be made very thin while still encapsulated, as well as withstand repeated bending at very small bending radius, which would be difficult if there is a plastic film substrate. I agree with the authors that there is no industry standard to quantify the WVTR. The 10-6g/m²/day of WVTR is beyond the capability of current measurement instruments. I accept the way the authors qualified their encapsulation property. So I am going to let the paper pass for publication.

Point-by-point reply

Reviewer #2 (Remarks to the Author):

With the satisfactory responses arising from the authors, my previous concern is completely resolved. I could find the other reviewers' comments and related authors' responses, in which there was the considerable conflict in between. Although both are definitely reasonable, I believe the more persuasive arguments are toward the authors' side. Therefore, I would like to strongly recommend the revised manuscript for publication in Nature Communications without further revision.

We thank Reviewer #2 for their positive appraisal of our manuscript and for recommending publication with further revisions.

Reviewer #3 (Remarks to the Author):

I recognize some of the authors' arguments. The industry do things differently from academics. They cannot compare like for like. The foldable phones have to pass much tougher tests in order not to upset the market and customers.

In term of innovations reported in this paper, putting an OLED film in water while it emitted light has been demonstrated by Holst Center years ago. The parylene/Al₂O₃ combination of multilayer barrier film was also reported before (Ref.36). The work which I think is innovative is to use the barrier film alone for OLED layers deposition. In this way, the OLED can be made very thin while still encapsulated, as well as withstand repeated bending at very small bending radius, which would be difficult if there is a plastic film substrate. I agree with the authors that there is no industry standard to quantify the WVTR. The 10⁻⁶g/m²/day of WVTR is beyond the capability of current measurement instruments. I accept the way the authors qualified their encapsulation property. So I am going to let the paper pass for publication.

We thank Reviewer #3 for their recommendation to pass our manuscript for publication. We agree that one of the main innovations is the use the barrier film alone for the deposition of the OLED layers. In our manuscript we argue and demonstrate that this configuration allows very thin, yet stable devices, well beyond what has been demonstrated before to our knowledge. We did not see any requests for further changes to our manuscript raised by Reviewer #3.